# Resistance to Thyroid Hormones: A Case-Series Study

**DOI:** 10.3390/ijms231911268

**Published:** 2022-09-24

**Authors:** Rossella Cannarella, Marco Musmeci, Vincenzo Garofalo, Tiziana A. Timpanaro, Guido Leone, Manuela Caruso, Paolo E. Maltese, Rosita A. Condorelli, Sandro La Vignera, Aldo E. Calogero

**Affiliations:** 1Endocrinology, Department of Clinical and Experimental Medicine, University of Catania, Via S. Sofia 78, 95123 Catania, Italy; 2Pediatric Endocrinology, Department of Clinical and Experimental Medicine, University of Catania, Via S. Sofia 78, 95123 Catania, Italy; 3MAGI Euregio, 39100 Bolzano, Italy

**Keywords:** resistance to thyroid hormone, RTH, Refetoff syndrome, thyroid hormones, total thyroidectomy, pregnancy, *THRβ*, metabolism, primary hyperparathyroidism

## Abstract

The aim of the study is to describe the clinical features of two unrelated patients with resistance to thyroid hormones (RTH), the first, a total thyroidectomized patient, and the second, a pregnant woman. We report the features found in her newborn who also showed RTH. Patient 1 is a 38-year-old man with total thyroidectomy managed for excessive thyroid stimulating hormone (TSH) production, which poorly responded to the replacement therapy. He was found with a *THRβ* c.1378G>A p.(Glu460Lys) heterozygous mutation, which was also present in other members of his family (son, brother, and father). Interestingly, Patient 1 had hypertension, dyslipidemia, and hepatic steatosis, which have been recently suggested as RTH-related comorbidities. Patient 2 is a 32-year-old pregnant woman with multinodular goiter, and the *THRβ* heterozygous variant c.959G>C, that, to the best of our knowledge, has been reported in literature only once. Her newborn had tachycardia and increased thyroid hormone levels, and showed the same mutation. After delivery, high parathyroid hormone (PTH) and calcium serum levels were found in Patient 2 and the scintigraphy showed the presence of adenoma of a parathyroid gland. This case-series study provides a practical example of the management of RTH in a thyroidectomized patient, a pregnant woman, and a newborn. A novel RTH pathogenic mutation is described for the second time in literature. Furthermore, the importance of metabolic assessment in patients with RTHβ has been highlighted and the possible correlation between RTH and primary hyperparathyroidism is discussed.

## 1. Introduction

Impaired sensitivity to thyroid hormone (ISTH) syndromes are a group of disorders characterized by an apparent discrepancy between serum levels of thyroid-stimulating hormone (TSH) and thyroid hormones (TH) and/or hormone action at the tissue and cell levels [1]. This group includes all defects that could interfere with the biological activity of properly synthesized THs. Defects in cell membrane transport, metabolism, and interaction with nuclear receptors have been identified to date [2]. ISTH can be subdivided into TH cell transport defect, in case of alterations of the cell membrane hormone transport proteins, and TH metabolism defect, when the anomaly is due to a non-physiological conversion of thyroxine (T4) into triiodothyronine (T3). ISTH can also be divided into abnormal hormone transfer to the nucleus and resistance to TH (RTH) [2].

RTH, a syndrome first described in 1967 by Samuel Refetoff [3], is mainly due to mutations of the *TH receptor β* (*THRβ*) gene and, less frequently, to mutations of the *TH receptor α* (*THRα*) gene [4]. These genes map to chromosomes 3 and 17, respectively. Each encodes two receptor isoforms: TR-α1, TR-α2, TR-β1, and TR-β2. They can all bind to THs except for TR-α2. In addition to varying at different stages of development, these receptor isoforms appear to have different functions and distribution throughout the body. TR-α1 is the predominant isoform in the heart, muscle tissue, and testis, whereas TR-β1 is mainly present in the brain, the liver, and kidneys. Furthermore, TR-β2 is mainly expressed in the hypothalamus, pituitary gland, inner ear, and retina [5]. To date, over 236 mutations of the *THRβ* gene mutations have been identified in patients with RTHβ [6] and all of these mutations are predominantly transmitted in an autosomal dominant manner.

The distinctive biochemical feature of RTHβ is the high serum levels of free-TH (fTH), while TSH concentrations are normal or elevated in the absence of other diseases that can cause these hormonal abnormalities [7]. The main presenting signs and symptoms in patients with RTHβ are goiter, short stature, attention deficit disorder, and resting tachycardia. However, some patients have goiter only or may be asymptomatic as the high levels of circulating TH can overcome tissue resistance [8]. Furthermore, patients with the same *THRβ* mutation can have different phenotypes, even within the same family [7]. These observations suggest that other genetic and epigenetic factors may play an important role in determining the RTHβ phenotype.

To better understand this rare disorder, it is useful to study as many cases as possible. In consideration of the heterogeneity of the phenotypes, the study of each case allows for the outlining of the best management of the patients with RTHβ. Furthermore, understanding the physiological adjustments of the hypothalamus-pituitary-thyroid axis occurring in this disease may be useful in uncovering important molecular mechanisms for feedback.

In this article, we present the cases of two unrelated RTHβ-affected families caused by two different mutations. The first case concerns a 38-year-old thyroidectomized patient managed by our outpatient clinic for elevated TSH values and then evaluated for mutations in the *THRβ* gene. After diagnosis, the analysis was extended to the whole family. The second case concerns a 39-year-old mother and her child suffering from hyperthyroidism and tachycardia at rest since birth. The genetic analysis made it possible to diagnose RTHβ in both mother and child.

## 2. Cases Presentation

### 2.1. Ethical Approval

Each participant was asked to sign an informed written consent, after a complete explanation of the aim of the study and of its procedures. The principles of the Declaration of Helsinki were strictly followed in the conduction of the study.

### 2.2. Bioinformatic and Genetic Analysis

An NGS analysis was performed on the blood samples of the enrolled patients. Sequencing was performed using a MiSeq personal sequencer (Illumina, San Diego, CA, USA). The methodology used for sequencing is detailed elsewhere [9]. International databases dbSNP (www.ncbi.nlm.nih. gov/SNP/, accessed on 1 March 2022) and Human Gene Mutation Database professional (HGMD; http://www.biobase-international.com/product/hgmd, accessed on 1 March 2022) were used for all nucleotide changes. The assessment of the pathogenicity of the chance of nucleotides was performed in silico using the Variant Effect Predictor tool (http://www.ensembl.org/Tools/VEP, accessed on 1 March 2022) and MutationTaster (http://www.mutationtaster.org, accessed on 1 March 2022). The Genome Aggregation Database (gnomAD) (http://gnomad.broadinstitute.org/, accessed on 1 March 2022) was used to check minor allele frequencies (MAF). The American College of Medical Genetics and Genomics guidelines [10] were followed for variants interpretation.

### 2.3. Patient 1

A 38-year-old patient was evaluated in the outpatient clinic of our center. To our knowledge, his medical history began in 2016, when he started taking ß-blockers to counteract tachycardia. To better understand the origin of this disorder, a study of the thyroid function was suggested by the Endocrinologist who was counseling the patient during that period. The diagnostic workup resulted in a picture of hyperthyroidism. Antithyroid antibodies were within the normal range. A subsequent ultrasound examination of the thyroid showed the presence of a multinodular goiter. He was counseled for a cytological evaluation of the thyroid resulting in TIR4, and, therefore, thyroidectomy was recommended. After surgery, ß-blockers were temporarily suspended and the patient began replacement therapy with L-thyroxine (100 µg/day). The histological analysis revealed the absence of thyroid cancer, and therefore he was prescribed substitutive therapy. Three months later, the abnormal serum TH levels led to increasing in the dose of L-thyroxin up to 228.6 µg/day. Despite several changes in L-thyroxine posology, serum TH levels could not be reported within an acceptable range in the following months. In 2018, the patient was referred to our Division of Endocrinology (University-Teaching Hospital, Policlinico “G. Rodolico—San Marco”, University of Catania, Catania, Italy). The patient had the following serum levels of TSH 20.56 µIU/mL (0.27–4.2), FT4 42.2 pmol/L (12–22), and FT3 7.22 pmol/L (3.1–6.8). As we started to reduce the daily dose of L-thyroxin, the patient recovered from tachycardia, and the treatment with ß-blockers was suspended. The levels of THs and L-thyroxine posology are summarized in Table 1. In 2019, a thyroid ultrasound (US) scan showed the presence of a hyperechoic oval area compatible with residual thyroid parenchyma, as confirmed by the scintigraphy that was not reported in the previous US scans. The patient was then suspected of RTH and underwent analysis of the *THRα* and *THRβ* genes by next-generation sequencing (NGS).

To exclude that the elevation of the serum THs was not due to pituitary causes, and since THR can also associate with pituitary adenomas [8], the patient underwent a magnetic resonance imaging (MRI) of the pituitary. The examination showed a normal pituitary gland.

Notably, the patient had extra thyroid comorbidities, such as hypertension (140/100 mmHg), dyslipidemia [total cholesterol 211 mg/dL, reference value (r.v.) < 200 mg/dL; HDL cholesterol 45 ng/dL, r.v. > 60 ng/dL; LDL cholesterol 164 mg/dL, r.v. < 130 ng/dL; triglycerides 207 mg/dL, r.v. < 150 ng/dL], hepatic steatosis, and *JAK-2* negative associated polycythemia, many of these being reported to be associated with *THRβ* gene mutations [11].

Gene analysis showed the presence in heterozygosity of the missense variant *THRβ* c.1378G>A p.(Glu460Lys), consisting in the presence of amino acid Glu460Lys in the THRβ protein. This variant concerns a highly conserved residue of the protein and alters its biochemical characteristics. It has already been reported in the ClinVar database in association with RTHß [12] and is interpreted by geneticists as pathogenic. Since this mutation is characterized by an autosomal dominant inheritance, the genetic investigation has been extended to other family members. We thus found that the patient’s father, brother, and son were also affected by the same variant of the *THRβ* gene (Figure 1). Their hormone serum levels are reported in Table 2.

Patient 1’s son was referred to have an Attention-Deficit / Hyperactivity Disorder (ADHD), which is known to be associated with THR [13]. We are not aware of any other disorder in the remaining family members.

### 2.4. Patient 2

The gynecologist of a 32-year-old pregnant patient requested counseling by our center for abnormal serum TH levels. Hormonal measurements were performed at the 13th and 16th weeks of gestation and showed an elevation of FT4 and FT3 associated with normal TSH values. Other measurements, performed at the 20th, 25th, and 27th weeks of gestation, showed an isolated increase in FT3 (Table 3). The patient had never performed tests of thyroid function before pregnancy.

An ultrasound scan of the patient’s thyroid showed a multinodular goiter. Thyroid scintigraphy showed both hot and cold nodules. Furthermore, Patient 2 showed high serum PTH and calcium levels, which led to the suspicion of primary hyperparathyroidism (PHPT). Parathyroid scintigraphy showed the presence of a hypercaptant area. Therefore, a diagnosis of parathyroid adenoma was made. No therapy was administered for neither the thyroid function nor for the hyperparathyroidism. The patient was counselled for parathyroidectomy, to be conducted after delivery.

The pregnancy ended with the natural delivery of a male newborn at the 37th week of gestation. Post-partum controls, on the other hand, showed hypocalcemia, thyrotoxicosis, and tachycardia, which required hospitalization at the Neonatal Intensive Care Unit (NICU) of our University-Teaching Hospital. Subsequent analysis of the newborn confirmed the diagnosis of hypocalcemia [calcium: 7.7 mg/dL, phosphorus: 1 mg/dL, and normal PTH, alkaline phosphatase (ALP), and 25-OH vitamin D levels] and thyrotoxicosis (Table 3). The ECG highlighted sinus tachycardia, although prenatal controls showed no abnormalities. These include the fetal echocardiography performed at the 21st week of gestation, which indicated a normal heart rhythm. Thyroid ultrasound showed that the gland was in situ and with regular morphology, diameters, and echo structure. No focal parathyroid lesions were observed.

The newborn was initially treated with calcium gluconate, 1α-hydroxycholecalciferol (as calcium levels did not respond adequately to cholecalciferol), and propranolol.

He was discharged from the NICU with the suspicion of RTH. Therefore, we suggested the genetic analysis of the *THRα* and the *TRHβ* genes by NGS. For this purpose, a blood sample was withdrawn from both the mother and the child. The DNA was then extracted and NGS performed. At 2 months and 4 months of age, calcium supplementation and treatment with propranolol were stopped, respectively. At 6 months, 1α-hydroxycholecalciferol was replaced with cholecalciferol. During the follow-up, we observed a progressive albeit slight decrease in TH levels up to 11 months of age, after which they stabilized. TSH was constantly in the normal range and no clinical signs of hyperthyroidism were present. Length and weight growth and psychomotor development were normal.

The genetic test showed the presence of the heterozygous variant c.959G>C, corresponding to the Arg320Pro amino acid substitution in the THRβ protein. This variant was found both in Patient 2 and in her son, and geneticists have interpreted it as pathogenic. As far as we know, it has been reported in the literature only once [14].

## 3. Discussion

RTH is a rare autosomal dominant or recessive inherited disease caused by mutation of the *THRα* or *THRβ* genes. The frequency of this syndrome varies among ethnic groups, with no difference between sexes. The precise incidence of RTH is unknown since neonatal screening based on TSH alone is not sufficient to diagnose it. However, two population studies measuring serum TSH and T4 levels on 80,884 and 74,992 newborns showed an incidence of 1 in 40,000 and 1 in 19,000 live births, respectively [15,16]. The most frequent causes are point mutations of the TR-β isoforms that cluster in three “hot spot” regions comprised from codons 234 to 282, codons 309 to 353, and codons 374 to 461 [4]. Not surprisingly, the mutations of the two patients reported in this study were found in codons 460 (patient 1) and 320 (patient 2). These hot spots are located in the coding regions for the T3 binding domain and the nearby hinge region. Therefore, these mutations alter the ability of the receptor to bind to its ligand as well as coactivators and to dissociate from corepressors [6]. To date, over 236 mutations of the *THRβ* in patients with RTHβ have been identified [6]. Almost all of these mutations are transmitted in an autosomal dominant manner because the mutated TRβ interferes with the function of the wild-type TRβ. However, it was not possible to identify any mutation in the *THRβ* gene in 14% of the patients with RTHβ. In some cases, this is due to mosaicism, while in others it has been postulated that there are mutations in the corepressor and coactivator genes [6].

The clinical manifestations of RTH vary widely among patients, making the disease phenotypically heterogeneous. The terms proposed in the past to classify RTH, such as generalized resistance to thyroid hormone, pituitary resistance to thyroid hormone, and peripheral tissue resistance to thyroid hormone, have now fallen into disuse. Indeed, patients with the same mutation, even belonging to the same family, often have different clinical manifestations. Goiter is the most common sign in patients with RTH, while other frequent symptoms are sinus tachycardia, short stature, developmental deficit, hair loss, frequent ear infections, and nose, throat, and psychological abnormalities. However, the clinical picture of patients can range from hyperthyroidism to hypothyroidism or the total absence of symptoms. In children, growth retardation, delayed bone maturation, and learning disabilities indicative of hypothyroidism may be present, together with hyperactivity and tachycardia, compatible with a picture of thyrotoxicosis.

Recently, the presence of metabolic abnormalities such as insulin resistance, dyslipidemia, endothelial dysfunction, and hepatic lipid accumulation has been reported in patients with THβ resistance. Therefore, careful surveillance should be reserved for these patients in the presence of these abnormalities [11]. In their clinical study, Moran and colleagues evaluated circulating lipids, intrahepatic and intramyocellular lipid content in an unselected cohort of adult RTHβ patients compared with age-, gender-, and body mass index (BMI)-matched healthy controls. The results showed an impaired level of circulating lipids (significantly higher LDL and triglycerides lipid levels, and significantly lower HDL levels), higher intrahepatic and intramyocellular lipids, HOMA index, and non-esterified fatty acids (NEFA) in patients than in controls. Therefore, according to the authors’ suggestions, lipid profile and hepatic liver content of RTHβ patients should be monitored to allow a timely prescription of lipid-lowering therapy in patients with hypercholesterolemia, or to recommend lifestyle changes (e.g., weight loss, physical activity) in patients with metabolic syndrome [11].

A previous study has shown that patients with a *THRβ* gene mutation have an increased content of intrahepatic fat compared to their normal relatives. Furthermore, while the intrahepatic fat content was directly associated with age and BMI in controls, this association was not found in RTHβ patients. Therefore, THR influenced the accumulation of fat more that age and obesity [17]. Likewise, an animal study using a mouse model with a mutation in the TRα gene, found a smaller liver and low lipids. These findings indicate a role for apo-TR isoforms in the pathogenesis of several features of hypothyroidism [18].

THs are involved in the regulation of hepatic lipid, cholesterol, and glucose metabolism. Therefore, their contribution in clinical conditions such as non-alcoholic fatty liver disease (NAFLD), type 2 diabetes mellitus, and insulin resistance, that are known to be linked to dysregulated hepatic metabolism, should not be excluded. Interestingly, THRβ expression negatively correlates with nonalcoholic steatohepatitis (NASH), serum triglyceride and HbA1c levels [19]. Intriguingly, the most highly dysregulated genes in the liver of NAFLD patients are those controlled by THs [20]. Increased accumulation of lipids in the liver enhances the expression of lipogenic enzyme genes, and reduces fatty acid oxidation in RTHβ transgenic mice [18]. In addition, some authors indicated the possible existence of a link between tyrosine kinase receptors [e.g., the insulin receptor (IR)], and the G protein-coupled receptors (e.g., the TSH receptor). Insulin resistance associates with an abnormal phosphorylation of IRS1, which, in turn, could interfere with the G protein-coupled receptors cascade. Altered IRS1 phosphorylation in T2DM patients and/or with insulin resistance may influence the TSHR signaling pathway, leading to higher serum TSH levels [21]. All these observations can explain the role of THs in lipid and glucose metabolism as modulators and activators of different genetic patterns and the importance of metabolic evaluation in patients with THRβ.

The TRß mutation found in Patient 2 and her son has been described in literature only once, in 31-year-old Japanese men presenting with tachycardia and elevated TH levels [14].

In addition, the patient also had PHPT, which may explain, through the suppression of fetal parathyroid gland, the hypocalcemia observed in the newborn. We wondered if there was a correlation between RTH and PHPT. In literature, a similar case was reported in Japan. The patient is a 57-year-old woman with elevated serum calcium, ALP, and parathyroid hormone accounting for PHPT. Furthermore, elevated FT4 and FT3 were also present with TSH levels in the normal range. A thyroid function test showed the presence of the syndrome of inappropriate secretion of TSH (SITSH) and, after careful examination, the diagnosis of RTH was made [22]. Several studies have examined the possible correlation between PHPT and thyroid diseases but no significant difference in the incidence of the former in different types of goiters was found [23,24,25]. Only non-medullary cancer of the thyroid was shown to be statistically more prevalent in PHPT than in autopsy controls [24]. The major factor that accounts for the coexistence of benign thyroid lesions and PHPT is that both are prevalent in middle-aged women [23,24]. Further analysis should be conducted in the future to evaluate whether a correlation between RTH and PHPT occurs and the possible mechanism(s) underlying this association.

Children with RTH may show goiter, emotional disturbances, delayed bone age, tachycardia, low BMI, attention deficit hyperactivity disorder, learning disability, hyperkinetic behavior, short stature, hearing loss (sensorineural), and mental retardation [26]. With the exception of tachycardia, none of these signs were observed in patient 2′s son, up to 21 months of age. Increased levels of both total and free THs and unsuppressed TSH (that may be normal or slightly increased) are concomitantly present in RTH. To distinguish between RTH and TSH-producing pituitary tumors may be challenging as THs and TSH concentrations are not different. On the other hand, pituitary imaging can help in the differential diagnosis [27]. Interestingly, while Patient 2 exhibited normal or marginally increased FT4 levels, her newborn showed a significant increase in FT4 levels. The reason for this discordance is not clear. A study published in 2017 reported that individuals exposed to high maternal TH levels in utero due to THR can develop central resistance to TH [28]. Whether the in utero exposure to high TH may have worsened the TH resistance in the newborn, thus contributing to the marked increase in the FT4 levels, is unknown.

The treatment depends on the specific features of each patient. Those in a compensated euthyroid state do not need any treatment. Patients with hypothyroid and hyperthyroid symptoms may require THs, or ß-blockers, anti-thyroid compounds, and thyroid hormone analogs [26]. Interestingly, the management of pregnancy in patients with RTH associated with Hashimoto’s thyroiditis is conservative. Accordingly, fT4 levels need to be maintained within a maximum of 20% above the upper limit of the reference range, independently from the presence of the mutation in the fetus [29]. Finally, it was shown that adult humans without RTHβ, exposed to high maternal THs in utero, showed persistent central resistance to THs as shown by the reduced response of serum TSH to TRH when treated with T3 [28]. Whether this mechanism concurred to the clinical findings described in the newborn reported in the present case-series study remains unknown.

## 4. Conclusions

RTH is a clinical syndrome defined by impaired sensitivity to THs and its more common form is caused by mutations in RTHβ. The biochemical profile includes elevated serum TH levels in the absence of TSH suppression. The cases herein reported highlights emerging areas of interest, such as the implications of RTHβ lipid and glucose metabolism. We also reported the clinical features as well as our experience in the management of RTHβ in a thyroidectomized patient, a pregnant woman, and her newborn. Furthermore, we herein describe, for the second time in literature, a novel pathogenic *RTHβ* gene variant. The advances in our knowledge of RTHβ raise novel questions about emerging concepts on RTHβ that need to be explored further.

## Figures and Tables

**Figure 1 ijms-23-11268-f001:**
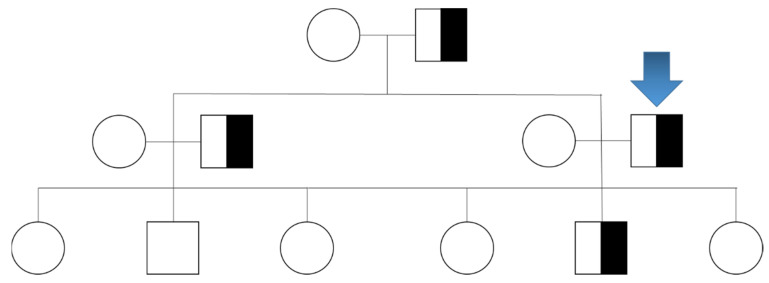
Mutation family tree. Patient 1 is indicated with a blue arrow, and shows the heterozygous *THRβ* c.1378G>A p.(Glu460Lys) mutation. The same mutation was found in his father, brother, and son. Circles indicate females, squares indicate males.

**Table 1 ijms-23-11268-t001:** Hormone serum levels and posology of L-thyroxine replacement therapy in Patient 1.

Date	TSH (µIU/mL)	FT4 (pmol/L)	FT3 (pmol/L)
September 2016	0.85	38.61	6.23
March 2017	Total thyroidectomy. Prescription of L-thyroxine 175 µg once daily		
July 2017	48.42	12.01	3.28
	Prescription of L-thyroxine 228.6 µg once daily		
November 2017	19.13	12.87	3.37
June 2018	16.83	12.87	3.58
October 2018	20.56	42.2	7.22
October 2018	Prescription of L-thyroxine 200 µg once daily		
October 2018	18.41	41.64	7.17
November 2018	22.74	30.96	7.67
December 2018	27.93	22.6	6.69
December 2018	Prescription of L-thyroxine 196.4 µg once daily		
February 2019	26.31	33.23	7.72
February 2019	Prescription of L-thyroxine 200 µg once daily		
March 2019	30.67	19.7	7.5
April 2019	Prescription of L-thyroxine 214.2 µg once daily		
May 2019	21.11	23.4	7.69
May 2019	Prescription of L-thyroxine 207.9 µg once daily		
September 2019	23.9	21.2	6.84
October 2019	Prescription of L-thyroxine 203.6 µg once daily		
September 2020	10.61	26.4	7.46
February 2021	9.24	32.1	6.4
**Reference values**	**0.27–4.19**	**7.9–14.4**	**3.32–7.45**

Abbreviations: TSH: thyroid-stimulating hormone; FT4: free thyroxine; FT3: free triiodothyronine.

**Table 2 ijms-23-11268-t002:** Hormone serum levels of the relatives of the Patient 1.

	TSH (µIU/mL)	FT4 (ng/dL)	FT3 (pmol/L)
**C.M.**			
June 2018	1.75	2.06	1.71
**C.Gi.**			
September 2016	2.06	2.12	7.01
**C.Ga.**			
September 2016	1.14	2.14	4.88
**Reference values**	**0.35–4.94**	**0.7–1.48**	**1.71–3.71**

Abbreviations: TSH: thyroid stimulating hormone; FT4: free thyroxine; FT3: free triiodothyronine. C.M., C.Gi., and C.Ga. are the father, the brother and the son of Patient 1, respectively.

**Table 3 ijms-23-11268-t003:** Hormonal findings in patient 2 and her newborn son.

	Patient 2				Newborn		
	Pregnancy Week	TSH (μIU/mL)	FT4 (ng/dL)	FT3 (pmol/L)	TSH (μIU/mL)	FT4 (ng/dL)	FT3 (pmol/L)
November 2020	13th + 1	1.02	1.5	5.55	---	---	---
May 2020	16th + 2	1.53	1.62	7.58	---	---	---
March 2020	20th + 3	1.36	1.43	6.32	---	---	---
May 2020	34th	1.06	1.25	4.43	---	---	---
March 2020	Delivery						
October 2020	---	0.72	1.91	4.51	6.62	41.25	---
November 2020	---	1.15	1.62	4.92	---	39.6	10.87
December 2020	---	---	---	---	3.99	34.2	9.8
January 2021	---	2.07	1.64	5.34	3.94	36.3	12.47
February 2021	---	---	---	---	3.66	32.1	10.03
March 2021	---	---	---	---	2.74	30.9	---
May 2021	---	1.14	1.66	4.54	4.51	26.3	---
September 2021	---	---	---	---	1.9	20.3	---
November 2021	---	---	---	---	---	---	---
March 2022	---	---	---	---	2.96	22.5	9.96
July 2022	---	---	---	---	3.27	26.9	9.93
**Reference values**		**0.35–4.94**	**0.7–1.48**	**1.71–3.71**	**0.34–4.2**	**6.8–16**	**3.8–6.0**

Abbreviations: TSH: thyroid-stimulating hormone; FT4: free thyroxine; FT3: free triiodothyronine.

## Data Availability

Data are available upon request to authors.

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
