# Peer review of "Resistance to Thyroid Hormones: A Case-Series Study"

_ijms, 2022, doi:10.3390/ijms231911268_

Round 1
Reviewer 1 Report
To the Authors:
General comments:
The authors investigated the clinical features of two unrelated patients with resistance to thyroid hormones (RTH). Also, they reported a new variant of THRβ. It was considered that the study was well written, and the result included novelty. However, several points should be addressed to improve the manuscript.
Specific comments:
1. Patient 1 was diagnosed with hyperthyroidism by the previous doctors. What were the serum levels of thyroglobulin and antithyroid antibodies in Patient 1?
2. Patient 1 was thyroidectomized since the cytological evaluation of the thyroid resulted in TIR4. What was the result of the pathological analysis?
3. Were the RTH and PHPT in Patient 2 treated? Please describe the treatment if they were treated.
4. Overall, more detail data and clinical characteristics of each patient and also other control and/or carrier cases would be disclosed.
Author Response
Manuscript ID: ijms-1933731
Comment 1. Patient 1 was diagnosed with hyperthyroidism by the previous doctors. What were the serum levels of thyroglobulin and antithyroid antibodies in Patient 1?
Answer to comment 1. Dear Reviewer, we thank you for the time spent in reviewing our manuscript, as well for your valuable comments. Both thyroglobulin and antithyroid antibodies were within the normal range.
Comment 2. Patient 1 was thyroidectomized since the cytological evaluation of the thyroid resulted in TIR4. What was the result of the pathological analysis?
Answer to comment 2. It was negative for thyroid cancer. We apologize for forgetting this important detail. This has now been added.
Comment 3. Were the RTH and PHPT in Patient 2 treated? Please describe the treatment if they were treated.
Answer to comment 3. No therapy was administered for neither the thyroid function nor for the hyperparathyroidism. The patient was counselled for parathyroidectomy, to be done after delivery.
Comment 4. Overall, more detail data and clinical characteristics of each patient and also other control and/or carrier cases would be disclosed.
Answer to comment 4. We have revised the clinical charts and included additional available clinical information. Particularly, we found that the son of Patient 1 was diagnosed with ADHD, which is known to be associated to THR. We are not aware of other diseases in the remaining members. Thank you.
Reviewer 2 Report
Dear Authors,
This manuscript describes three different and interesting clinical cases of resistance to thyroid hormones. This is an infrequent and heterogeneous entity which requires a high level of suspicion to identify properly. The gathering of these three cases is certainly valuable.
The introduction contains a clear explanation of the pathogenesis and metabolic disturbances associated to this group of genetic diseases. The cases have been presented in a correct way and the discussion is pertinent and comprehensive.
Minor comments:
a) In patient 1: how do you explain the treatment with levothyroxine from October-2018 to October-2019 meanwhile the patient showed FT4 levels permanently out of the normal range??
b) Table 2: It is supposed that C.M; C.Gi, and C.Ga are the different relatives, but this is not clarified at the table.
c) Patient 2-patient 3: meanwhile the mother exhibited normal or marginally increased FT4 levels, her newborn showed a significant increase in FT4 levels. This discordance mother-baby should be highlighted at the discussion.
Author Response
Manuscript ID: ijms-1933731
Comment 1. In patient 1: how do you explain the treatment with levothyroxine from October-2018 to October-2019 meanwhile the patient showed FT4 levels permanently out of the normal range??
Answer to comment 1. The patient was thyroidectomized at that time, and he needed hormonal replacement therapy. We tried to bring TSH in acceptable range by gradually reducing the dosage. The permanently high FT4 have to be ascribed to the hormonal resistance.
Comment 2. Table 2: It is supposed that C.M; C.Gi, and C.Ga are the different relatives, but this is not clarified at the table.
Answer to comment 2. This has been added (please see table 2). Thank you.
Comment 3. Patient 2-patient 3: meanwhile the mother exhibited normal or marginally increased FT4 levels, her newborn showed a significant increase in FT4 levels. This discordance mother-baby should be highlighted at the discussion.
Answer to comment 3. Thank you for you comment. We added the following period:
“Interestingly, while Patient 2 exhibited normal or marginally increased FT4 levels, her newborn showed a significant increase in FT4 levels. The reason for this discordance is not clear. A study published in 2017 reported that individuals exposed to high maternal TH levels in utero due to THR can develop central resistance to TH (Srichomkwun et al., 2017). If the in-utero exposure to high TH may have worsened the TH resistance in the newborn, thus contributing to the marked increase in the FT4 levels, is unknow”.
Round 2
Reviewer 1 Report
To the Authors:
General comments:
The authors revised the manuscript according to the comments.